# Study on the Failure Mechanism of a Modified Hydrophilic Polyurethane Material Pisha Sandstone System under Dry–Wet Cycles

**DOI:** 10.3390/polym14224837

**Published:** 2022-11-10

**Authors:** Wenbo Ma, Peng Tang, Xuan Zhou, Guodong Li, Wei Zhu

**Affiliations:** 1School of Mechanical Engineering and Mechanics, Xiangtan University, Xiangtan 411105, China; 2School of Mathematics and Computational Science, Xiangtan University, Xiangtan 411105, China

**Keywords:** Pisha sandstone, dry–wet cycles, W-OH, failure mechanism

## Abstract

Modified hydrophilic polyurethane is a new and effective material for soil and water conservation, which can form a consolidation layer with soil and has achieved more outstanding results in slope management in Pisha sandstone areas. However, the W-OH–Pisha sandstone system can be destroyed by local natural conditions, which seriously affects its consolidation effect on the soil. This paper focuses on the failure mechanism of the W-OH–Pisha sandstone system under dry–wet cycles; it establishes its failure model and provides theoretical guidance on the use of W-OH materials for slope management. Firstly, mechanical and in-situ morphological observations of W-OH solid consolidation under dry–wet cycles were carried out, and the results showed that W-OH solid consolidation at different concentrations only becomes rougher, and their cohesive failure does not occur under dry–wet cycles. Then, the adhesion model and water damage model of the W-OH–Pisha sandstone system were established based on surface energy theory. It was found that the larger the concentration of W-OH, the better the adhesion and spalling resistance performance. Additionally, we used the water stability constant to express the compatibility of W-OH with Pisha sandstones. The results showed that the greater the concentration of W-OH, the greater the water stability constant and the resistance of the W-OH–Pisha sandstone solid consolidation to the dry–wet cycles. Finally, based on the unconfined compressive strength test of the W-OH–Pisha sandstone solid consolidation, the unconfined strength ratio of the W-OH–Pisha sandstone solid consolidation was established as a function of the water stability constant; the unconfined strength ratio of the solid consolidations increases with an increase in the water stability constant. This also verifies the correctness of the W-OH–Pisha sandstone adhesion model and the water damage model.

## 1. Introduction

The Pisha sandstone is a loose rock series consisting of sandstone, sand shale, and argillaceous sandstone. It is widely distributed in the Shaanxi Province and the Inner Mongolia Autonomous Region in the Yellow River Basin of China, covering an area of about 16.7 × 10^3^ km [1,2,3]. The mineral composition of the Pisha sandstone is quartz, montmorillonite, feldspar, and calcite. The feldspar, for which the main weathering material is kaolinite, is poorly resistant to erosion and is the main mineral responsible for the susceptibility of the Pisha sandstone to erosion [4]. Montmorillonite is extremely swellable when exposed to water, with a maximum swelling rate of 150%, and is the main reason why the Pisha sandstone is susceptible to collapse when exposed to water [5]. As a continental clastic series, the small thickness and low pressure of the overlying rock layers result in a low degree of diagenesis, a low degree of intersand cementation, and low structural strength [6,7]. In addition, due to the special lithology of the Pisha sandstone, which is extremely sensitive to water, it becomes mud when water is encountered, causing serious erosion problems in the area [8,9]. The slope is an important area where sediment is generated during erosion [10]. This also caused frequent instability of the slope soil in the Pisha sandstone area. 

In order to solve the problem of slope instability in Pisha sandstone areas, numerous scholars have carried out lots of studies. Zaslavsky [11] developed a graft copolymerization of lignin sulfate with vinyl monomers for the preparation of soil conditioners. You et al. [12] studied the effect of sea buckthorn forests on flood control and sand reduction. The first application of the EN-1 curing agent for the treatment of the Pisha sandstone area was at NWAS [13]. Gautam et al. [14] analyzed the thermodynamic properties of different types of sandstones at high temperatures. Eventually, great breakthroughs in materials were made. Modified hydrophilic polyurethane material (W-OH) is a new type of soil stabilizer that reacts quickly with water to release CO_2_ gas and forms a porous flexible gel with a certain chemical strength, which effectively permeates into the weathered Pisha sandstone particles with water; after curing, loose soil can connect with the retention fertilizer effect of large mesh structures, having a good effect of consolidation. For example, some people applied W-OH to sand fixation and found that W-OH can form a flexible sand fixation layer with certain mechanical properties and porosity, which can not only resist wind and hydraulic erosion but also significantly alleviates the volume change caused by frost swelling and gravitational forces [15,16]. In fact, early studies have found that polymers can strengthen soils, reduce permeability, and control soil erosion [17,18,19]. After that, scholars conducted a lot of research on soil reinforcement based on the W-OH material [20,21,22]. Liang et al. [23] proposed a type of W-OH material to control its erosion; the consolidation layer it forms can reduce the rate of water erosion and reduces evaporation. Wang et al. [24] used polyurethane materials to reinforce the silt embankment and improved the compaction and mechanical properties of the embankment. Currently, it is internationally accepted that the use of W-OH materials for slope protection in Pisha sandstone areas is proven to be a cost-effective slope protection measure.

However, so far, all domestic and international studies related to W-OH have largely applied its consolidation properties to soil reinforcement, almost without considering the failure of W-OH solid consolidation under the action of the natural environment, and studies on its failure mechanism are also scarce. In addition, considering that W-OH adheres to the surface of Pisha sandstone particles after solidification and acts as a bridge between the Pisha sandstone particles, W-OH is essential for controlling soil erosion in Pisha sandstone areas, so it is significant to study the failure characteristics of the W-OH–Pisha sandstone system.

Fortunately, as our group’s research into the consolidation of Pisha sandstones based on W-OH materials progressed, it was found that the consolidation layer formed by spraying W-OH on the surface of the Pisha sandstone can be destroyed under the influence of local natural conditions (dry–wet cycles and freeze–thaw cycles) [25,26]. The solid consolidations wrapped around the surface of the Pisha sandstone are affected by some external factors; different degrees of damage occurs between the interface of the Pisha sandstone particles and W-OH solid consolidations, and the phenomenon of shedding occurs, which affects the sand consolidation effect of the W-OH material, with the damage aggravated over time. However, it is difficult to explain this phenomenon, as there is little discussion about the failure mechanism of the W-OH–Pisha sandstone system. After extensive studies, it was found that there is a high degree of similarity between the failure between the asphalt–aggregate system and the W-OH–Pisha sandstone system. Xu et al. [27] used studies of the contact angle and the surface free energy to characterize the adhesion between asphalt and aggregates in a saline environment. Zhou et al. [28] quantitatively analyzed the roughness of the aggregate and the contact angle of the asphalt on the aggregate surface. Molecular dynamics simulations were carried out to evaluate the adhesion properties between the aggregate and the asphalt, molecular diffusion, and the interface failure mechanism. Hefer et al. [29] measured the contact angle between the aggregate and the asphalt by the hanging piece method and calculated the surface energy to select the aggregate and the asphalt. Bhasin et al. [30] estimated the adhesion work between the asphalt and the aggregate using a microcalorimeter and believed that using the microcalorimeter method presents a quick and easy way to estimate the surface energy. Yuan et al. [31] proposed the use of the difference between the contact angle of the asphalt and the aggregate and the contact angle of the water and the aggregate to express the water stability of the asphalt mixture. Zheng et al. [32] calculated the adhesion work of the asphalt and the aggregate under different conditions by determining the surface energy of the asphalt and the aggregate, and the results of the study were consistent with adhesion theory, indicating that the surface energy theory is feasible for use in the evaluation of the adhesion of asphalt mixes with their water stability. After feasibility considerations, we envisaged whether surface energy theory could be used to derive the failure mechanism of the W-OH–Pisha sandstone system, with reference to the asphalt–aggregate system. Since there are no previous studies on the failure characteristics of the W-OH–Pisha sandstone system, the use of this method will represent a pioneering study.

This paper focuses on the failure mechanism of the W-OH–Pisha sandstone system under dry–wet cycles; the failure type is analyzed, and the adhesion model and water damage model are derived, providing a theoretical guide for the treatment of Pisha sandstone slopes based on W-OH materials. Firstly, the mechanical properties of the W-OH solid consolidation were analyzed based on the Instron universal testing machine, and the surface morphology of the W-OH solid consolidation of different concentrations under dry–wet cycles was analyzed by using metallographic microscopy, specifically using the binarization method for processing the images to investigate the failure characteristics of the W-OH solid consolidation itself, under dry–wet cycles. Then, surface energy theory was applied to the damage model, and, based on the adhesion process and the water damage process of the W-OH–Pisha sandstone system, the adhesion model and the water damage model of W-OH–Pisha sandstone system was established, and their water stability constants were obtained through the energy change ratio of the water damage process to the adhesion process, which can be used to judge the resistance of W-OH–Pisha sandstone solid consolidation with different concentrations against dry–wet cycles. Finally, the unconfined strength ratio of W-OH–Pisha sandstone solid consolidation, before and after the dry–wet cycles, can be expressed as a function of the water stability constant through the unconfined compressive strength test of W-OH–Pisha sandstone solid consolidation to judge the correctness of the established model.

## 2. Materials and Methods

### 2.1. Test Material

The W-OH material used in this paper was obtained by the polymerization of isocyanate, polyether polyol, and various functionally modified materials under specific temperatures, times, and ratio conditions based on the original hydrophilic polyurethane material incorporating nanomodification, composition structure change, and functional material composite technology. The main materials used in the synthesis of W-OH are shown in Table 1, and the standards of the materials were all industrial level. Moreover, when the concentration of W-OH is less than 4%, it has a good growth-promoting effect on the plants of the slope; when the material concentration is higher than 4%, it has a good consolidation effect on the slope. 

The Pisha sandstone samples were taken from the Erhuogou basin in the Huafuchuan basin (Ordos City, China), a primary tributary of the Yellow River, where the Pisha sandstone is mainly greyish-white, loose in structure, and severely fractured by water, and is generally a more severely weathered Pisha sandstone. In order to obtain the mineral composition of the Pisha sandstone sample, an XRD (Bruker, Bremen, Germany) test was carried out on the Pisha sandstone sample. Firstly, the sample was oven dried at 105 °C for 24 h; then, the dried sample was ground to obtain the powdered sample with a particle size of fewer than 80 μm. Afterward, the sample was scanned using a D8-Discover X-ray diffractometer, for which the detector type was LynxEte, and its X-ray diffraction pattern was obtained. The test was set to a two-theta range of 0–90°, with a step width of 0.02° and a scanning speed of 0.15 s/step. Finally, in combination with the X-ray diffraction pattern, the standard powder diffraction data for the various substances provided by the Powder Diffraction Consortium International Data Centre (JCPDS-IC-DD) were used for comparative analysis. Then, the mineral composition of the Pisha sandstone sample was determined, and the relative content of each phase was also quantified. The test results are shown in Figure 1. It was found that the sample contains 40.26% quartz, followed by 29.61% feldspar, 21.93% montmorillonite, and 5.93% calcite, with small amounts of chlorite and mica at 1.19% and 1.08%, respectively.

### 2.2. Experimental Design

#### 2.2.1. Preparation of the W-OH Solid Consolidation Samples

The manufacturing method of W-OH solid consolidation is described as follows: firstly, the acrylic plate container with a volume of 10 × 10 × 0.5 cm^3^ was made. Then, W-OH solutions of different concentrations were prepared, with the water and W-OH fully stirred and poured into the container. Finally, after consolidation, we placed the sample in a drying box and left it to consolidate into a film for use. The detailed data are shown in Table 2. Figure 2 shows the flow chart of the preparation process of W-OH solid consolidation, and Figure 3 shows W-OH solid consolidations using different concentrations.

#### 2.2.2. Dry–Wet Cycle Method

The dry–wet cycle process was as follows: (1) prepare the finished W-OH solid consolidations using different concentrations; (2) put the W-OH solid consolidations in water at room temperature for 12 h; (3) put those W-OH solid consolidations that had completed the wetting process into an oven at 45 °C for 12 h. During the process of drying, the surface of the solid consolidation is covered with filter paper to prevent the rest of the material from damaging the surface during the drying process. (4) Take them out with forceps and observe. Steps (1)–(4) are considered a dry–wet cycle. 

#### 2.2.3. Tests of Mechanical Properties of the W-OH Solid Consolidation under Dry–Wet Cycles 

The tensile test of the W-OH solid consolidation was carried out based on an Instron universal testing machine (Shanghai Hualong Test Instruments Co., Ltd., Shanghai, China). The tensile sample is a long strip with a length of 4 cm and a width of 0.5 cm, and the actual tensile length is 2 cm.

The fracture strength of the solid consolidation can be calculated by Equation (1):(1)$$\sigma_{c}=\frac{F}{ah}$$
where *F* is the test machine reading (N); a is the thickness of the solid consolidation; and *h* is the width of the sample, which is 5 mm, as is shown in Table 3.

#### 2.2.4. In-Situ Morphological Observation Tests of the W-OH Solid Consolidations under Dry–Wet Cycles 

In this experiment, the Olympus metallographic microscope (Shanghai Puhe Photoelectric Technology Co., Ltd., Shanghai, China) was used to observe the solid consolidation after the dry–wet cycles of W-OH using concentrations of 4%, 6%, 8%, and 10%, respectively. The method of the in-situ observation is as follows: firstly, the solid consolidation is cut to a size of about 2 cm, and a cross is drawn on it with a marker pen, with the numbers 1, 2, 3, and 4 written onto the four quadrants formed by the cross; finally, place it on a glass slide and observe the four positions of the cross through a microscope, as is shown in Figure 4.

#### 2.2.5. Theoretical Derivation of Surface Energy

Surface energy theory suggests that the adhesion between the liquid and the solid comes from the wetting of the solid surface by the liquid, a process that reduces the surface free energy of the system. There is an unsaturated force field on the surface of the solid, so it will spontaneously attract other substances to reduce its free energy. When W-OH diffuses and wets the surface of Pisha sandstone, the particles spontaneously absorb the molecules of the W-OH solution to reduce the surface free energy, and adhesion is formed as a result of this energy effect. Besides, the surface tension of the liquid and the solid can be expressed as the sum of the dispersion component and the polar component. The dispersive component is the very weak attractive force created when nonpolar molecules come close to each other and form the nonpolar part of the van der Waals force, while the polar component consists of dipole forces, induced forces, and hydrogen bonds. The surface energy equation is given in Equation (2):(2)$$\gamma =\gamma^{d}+\gamma^{p}$$
where γ is the surface energy (mJ/m^2^); $\gamma^{d}$ is the dispersion component (mJ/m^2^); and $\gamma^{p}$ is the polar component (mJ/m^2^). 

The surface free energy of the solid–liquid interface is expressed in Equation (3): (3)$$\gamma_{SL}=\gamma_{S}+\gamma_{L}-2\sqrt{\gamma_{S}^{d}\gamma_{L}^{d}}-2\sqrt{\gamma_{S}^{p}\gamma_{L}^{p}}$$
where $\gamma_{SL}$ is the surface free energy of the solid–liquid interface (mJ/m^2^); $\gamma_{S}$ is the free energy of the solid surface (mJ/m^2^); $\gamma_{L}$ is the free energy of the liquid surface (mJ/m^2^); $\gamma_{S}^{d}$ is the dispersion component of the solid (mJ/m^2^); $\gamma_{S}^{p}$ is the polar component of the solid (mJ/m^2^); $\gamma_{L}^{d}$ is the dispersion component of the liquid (mJ/m^2^); and $\gamma_{L}^{p}$ is the polar component of the liquid (mJ/m^2^). 

Combining with Yong’s [33] equation: (4)$$\gamma_{S}=\gamma_{SL}+\gamma_{L}\cos\theta$$

Equation (5) can be obtained: (5)$$\gamma_{L}(1+\cos\theta )=2\sqrt{\gamma_{S}^{d}\gamma_{L}^{d}}+2\sqrt{\gamma_{S}^{p}\gamma_{S}^{p}}$$

According to surface energy theory, the adhesion work can characterize the adhesion strength of the solid–liquid interface. When the W-OH solution is sprayed onto the Pisha sandstone, it is equivalent to the wetting effect of the liquid on the solid. When equilibrium is reached, a contact angle develops between the solid and the liquid. The contact angle can be measured directly and used to calculate the adhesion work. The expression for the adhesion work is shown in Equation (6):(6)$$W_{SL}=\gamma_{L}+\gamma_{S}-\gamma_{LS}$$
where $W_{SL}$ is the adhesion work, and the meanings of the other symbols are the same as above. 

Substituting Equation (3) into Equation (6), we can get:(7)$$W_{SL}=2\sqrt{\gamma_{S}^{d}\gamma_{L}^{d}}+2\sqrt{\gamma_{S}^{p}\gamma_{S}^{p}}=\gamma_{L}(1+\cos\theta )$$
and the change of the surface free energy is shown in Equation (8):(8)$$\Delta G_{SL}=-W_{S}$$

Therefore, it is only necessary to substitute the contact angle between the solid and the liquid and the measured value of the surface free energy of the liquid into Equation (7) to calculate the adhesion work and the change value of the surface free energy between the solid and the liquid. 

Equation (7) calculates the adhesion property of the Pisha sandstone particles and the W-OH solid consolidations under a nonwater condition. However, in practical engineering, the W-OH–Pisha sandstone system is bound to suffer from the effects of rainwater, where dry–wet cycles repeatedly occur. The W-OH–Pisha sandstone solid consolidation undergoes exfoliation damage due to poor adhesion, and water gradually replaces the film of the W-OH solid consolidation, and, eventually, the surface of the Pisha sandstone particles is wrapped in a water film. The process can be expressed as:


W-OH–Pisha sandstone + Water→Pisha sandstone − Water + W-OH-Water


The W-OH–Pisha sandstone system will form two systems, including the Pisha sandstone-water system and the W-OH-water system under the action of water. The relationship between the adhesion of the W-OH–Pisha sandstone system and its surface free energy is shown in Equation (9):(9)$$\Delta G_{SLW}=\gamma_{LW}+\gamma_{sW}-\gamma_{SL}$$
where $\gamma_{LW}$, $\gamma_{sW}$, and $\gamma_{SL}$, respectively, represent the interface energy of W-OH and water, Pisha sandstone, and W-OH and Pisha sandstone. 

According to Equations (3) and (5), this can be written as:(10)$$\begin{array}{ll} \Delta G_{SLW} & =\gamma_{L}+\gamma_{W}-2\sqrt{\gamma_{L}^{p}\gamma_{W}^{p}}-2\sqrt{\gamma_{L}^{d}\gamma_{W}^{d}}+\gamma_{S}+\gamma_{W}-2\sqrt{\gamma_{S}^{p}\gamma_{W}^{p}}-2\sqrt{\gamma_{S}^{d}\gamma_{W}^{d}}-\gamma_{S}-\gamma_{L}+2\sqrt{\gamma_{S}^{p}\gamma_{L}^{p}}+2\sqrt{\gamma_{S}^{d}\gamma_{L}^{d}}\\ & =2\gamma_{W}+\gamma_{L}\left(1+\cos\theta_{SL}\right)-\gamma_{W}\left(1+\cos\theta_{LW}\right)-\gamma_{W}\left(1+\cos\theta_{SW}\right) \end{array}$$
where $\gamma_{W}$ represents the surface tension of water, with a value of 72.8 ${\mathrm{mJ}/m}^{2}$; $\gamma_{L}$ represents the surface energy of W-OH; $\theta_{SL}$, $\theta_{LW}$, and $\theta_{SW}$, respectively, represent the contact angle of W-OH and Pisha sandstone, water and W-OH, and water and Pisha sandstone. The greater the absolute value of $\Delta G_{SLW}$, the deeper the exfoliation, and, at the same time, the more W-OH material will fall off from the Pisha sandstone. 

#### 2.2.6. Droplet Contact Angle Tests of W-OH Solid Consolidations under Dry–Wet Cycles 

The contact angle was measured by a JC2000D1 contact angle measuring instrument produced by Shanghai Zhongchen Digital Technology Equipment Co., Ltd., (Shanghai, China). The contact angle test method in this chapter is the lying drop method. In the test, a series of drops of liquid with a known surface energy is dropped onto the solid surface, and the shape of the drops formed by the liquid and the solid plane is described by the contact angle; the exact contact angle is the angle at the junction of the solid, liquid, and gas phases from the solid–liquid interface through the liquid interior to the gas interface. This method is a commonly used method for measuring the contact angle. By using this method, two correct contact angles can be obtained simultaneously, which are called the left contact angle and the right contact angle. Generally speaking, the two contact angles are basically the same.

The test liquids were chosen to be ethylene glycol and formamide (Shanghai Qinba Chemical Co., Ltd., Shanghai, China) because they have a large surface free energy and are not soluble with W-OH solids and have different surface energy fractions. The surface free energy fraction of the W-OH solids can be obtained from Equation (5), and the surface free energy of the two liquids and their fractional values are shown in Table 4, where both ethylene glycol and formamide are 99% analytically pure.

#### 2.2.7. Failure Tests of the W-OH–Pisha Sandstone Solid Consolidation under Dry–Wet Cycles

A CMT5504/5105 universal testing machine was used for the test. The samples were fixed to the machine, and an axial pressure was applied to the sample at a controlled rate of 0.8 mm/min while the machine recorded the axial stress, *F*. In the test, the W-OH–Pisha sandstone solid consolidations without dry–wet cycles and after nine dry–wet cycles of different concentrations (4%, 6%, 8%, and 10%) were tested. Vaseline was applied to the contact surface of the sample with the lifting platform to increase lubrication and to avoid the horizontal restraint of the specimen on the platform. The load of the test curve has a maximum value, and the compressive stress at this maximum load is considered to be the unconfined compressive strength. The unconfined compressive strength is given by Equation (11):(11)$$\sigma_{c}=\frac{F}{A}$$
where *F* is the maximum load in the displacement-load curve during the test, and *A* is the cross-sectional area of the sample.

## 3. Results and Discussion

### 3.1. Study on the Cohesive Failure of the W-OH Solid Consolidations under Dry–Wet Cycles

#### 3.1.1. Mechanical Properties of the W-OH Solid Consolidations under Dry–Wet Cycles

Figure 5 shows the localization shift strength for each concentration (4%, 6%, 8%, and 10%). It can be found that the localization shift strength of the solid consolidations for each concentration fluctuates roughly within a range during the dry–wet cycles. This is because the effect of water on the solid consolidations during water immersion is mainly plasticizing. Water molecules enter into the macromolecular chain to form hydrogen bonds with the polar groups of polymer molecules, making the intermolecular forces in the polymer chain weaken, with the mechanical properties reduced. However, the process is reversible and can be restored to its original properties through the drying process of dehydration. In addition, since the solid consolidation is polyether-type polyurethane, the hydrolysis is very weak, so the dry–wet effect has little effect on it. The localization shift strength at this moment mainly depends on the pores and the number of molecules per unit volume of the solid consolidation. The smaller the pores, the more molecules per unit volume, and the greater the strength. At concentrations of 4 and 6%, the number of molecules per unit volume has a greater effect on the strength of the solid consolidation compared to the pores because there are fewer pores inside the solids. As the concentration increases, the number of molecules per unit volume of the solid consolidation increases, so the strength of the solid consolidation at a concentration of 6% is greater than that of 4%. At concentrations of 8 and 10%, the pores have a greater effect on the strength of the solid consolidation compared to the number of molecules per unit volume because there are more pores inside the solid consolidation; so much so that the strength of the 8% concentration solid consolidation is even lower than that of the 4% concentration solid consolidation. As the concentration increases, the pores of the solid consolidation increase. Therefore, the strength of the solid consolidation at 8% is greater than that of 10%. The average values of the localization shift strength for each dry–wet cycle (of 4%, 6%, 8%, and 10% concentrations) were 1.397, 1.945, 1.167, and 0.848 MPa, respectively. The fluctuation of the localization shift strength during the whole dry–wet cycle is due to a fluctuation in the pores inside the solid consolidations. However, the same concentration of solid consolidations with porosity (in a range) makes the localization shift strength fluctuate in a range. Therefore, the mechanical properties of the solid consolidations are not weakened by the dry–wet cycles, and the W-OH solid consolidations have excellent resistance to dry–wet cycles at all concentrations.

#### 3.1.2. In-Situ Observation of the Surface Morphology of the W-OH Solid Consolidation under Dry–Wet Cycles

The surface morphology of the W-OH solid consolidations for each concentration (4%, 6%, 8%, and 10%) under dry–wet cycles is shown in Figure 6. It was found that the number of dry–wet cycles has different effects on the solid consolidation for each concentration. 

Figure 6a shows that for the 4% concentration solid consolidation, the first dry–wet cycle produces a more pronounced change to it, with the appearance of grooves that were not previously present and also a more pronounced increase in roughness. This is due to the dark spots on the surface of the sample due to the reaction with the components in water and air, generating products such as alcohols, carboxylic acids, and esters. Thereafter, with an increase in the number of cycles, the dark spots increase dramatically, and the surface is rougher. When the number of dry–wet cycles is 12, it is no longer possible to find the original position because of too many dark spots on the surface. Figure 6b shows that, after the first dry–wet cycle, some grooves on the surface of the 6% concentration solid consolidation disappear, and the roughness decreases instead. This is because the solid consolidation will have some rearrangement of its molecules during the dry–wet cycles, which makes the surface flatter. However, as the number of cycles continues to increase, the surface of the solid consolidation will continue to produce dark spots and grooves. When the number of dry–wet cycles is 12, more dark spots are produced, but to a significantly lesser extent than with a concentration of 4%. Figure 6c,d show that the solid consolidations from the 8 and 10% concentrations do not produce many changes in their surface during the first three dry–wet cycles. This is because the surface is flatter due to the rearrangement of molecules. In contrast, after the fifth dry–wet cycle, the surface starts to change greatly. Thereafter, as the number of cycles continues to increase, dark spots and grooves continue to be produced, but at a lower rate when compared to the 4% and 6% concentrations. When the number of dry–wet cycles is 12, it is observed that the solid consolidation still has more areas without dark spots.

To quantify the surface morphological changes and the development of dark spots on the W-OH solid consolidations at different concentrations for each number of dry–wet cycles, the same area of the image shown above is binarized, as shown in Figure 7. Binarization involves setting the grey scale values of the pixel points in the image to 0 or 255 so that the whole image appears only distinctly black and white. Almost all of the current optical observations are made by the scanning electron microscope (SEM). However, in this paper, for the observations of the changes in the surface morphology of the solid consolidations, accurate results could be obtained just by using the binarization process. Dark areas are shown as black with an area of *S_B_*, and nondark areas are shown as white with an area of *S_W_*. The change in surface morphology is indicated by the dark spot rate, *k*, which is shown in Equation (12).
(12)$$k=\frac{S_{B}}{S_{B}+S_{W}}$$

The dark spot rate for the solid consolidation at each concentration for different numbers of dry–wet cycles is shown in Table 5. It was found that, for the 4% solid consolidations, the first dry–wet cycle increases the dark spot rate significantly, at 9.77%. After that, the dark spot rate increases sharply with an increase in the number of cycles until the ninth dry–wet cycle, when the dark spot rate reaches 70.2%. For the 6% solid consolidations, the first dry–wet cycle, instead, decreases the dark spot rate. Thereafter, the dark spot rate increases as the number of dry–wet cycles increase. When the number of dry–wet cycles is 12, the dark spot rate of the solid consolidation is 61.2%. For the 8 and 10% high-concentration solid consolidations, it was found that the first three dry–wet cycles do not have a significant effect on the surface, and the difference in dark spot rate is not significant. After the fifth dry–wet cycle, the dark spot rate started to increase, but at a much lower rate than was the case with the 4% and 6% concentrations. After the ninth dry–wet cycle, the dark spot rate for the two concentrations was found to change slowly, at around 2%. The dark spot rates for the 8 and 10% solid consolidations after 12 dry–wet cycles were 44% and 49.13% respectively, which are much lower than the dark spot rates at 4% and 6% concentrations. Therefore, for the W-OH solid consolidation, when the concentration is below 8%, the increase in resistance to the dry–wet cycles is more obvious as the concentration increases. While at concentrations above 10%, the resistance to the dry–wet cycles increase with increasing concentration, but the enhancement is not significant.

After studying the surface morphology and mechanical properties of the W-OH solid consolidations under dry–wet cycles, it was found that dry–wet cycles do not produce structural damage to the interior of the solid consolidation but only make the surface of the solid consolidation rough. It is believed that in the W-OH–Pisha sandstone solid consolidation, any cohesive damage caused by the lack of resistance to dry–wet cycles is difficult to corroborate. In addition, the adhesion force between the solid consolidations and the Pisha sandstone may change due to the roughness of the solid consolidation surface, resulting in the loss of W-OH from the surface of the Pisha sandstone and, thus, the adhesion damage.

### 3.2. Study on the Adhesion Failure of the W-OH Solid Consolidation under Dry–Wet Cycles

#### 3.2.1. Droplet Contact Angle Analysis of the W-OH Solid Consolidation under Dry–Wet Cycles

The left and right contact angles at 10 s, 30 s, and 60 s were obtained using analysis software. A total of five different locations were selected for each liquid on the surface of the W-OH solid consolidation, and the final average value was taken as the result. That is, the final contact angle was taken as the average of the 30 contact angle values. The surface energy parameters of the W-OH solid consolidation for each concentration were obtained by substituting the contact angle data and the surface free energy parameters of the test liquid into Equation (5). The relationship curves between the surface energy parameters and W-OH concentrations are shown in Figure 8. It was found that, as the concentration increases, the polar component of the solid consolidation at each concentration increases. This is mainly because the increase in concentration contributes to the generation of polar chemical bonds, such as carbamate and urea bonds, which leads to an increase in the intermolecular hydrogen-bonding forces, increasing the polar fraction [34,35]. While the dispersion component decreases with increasing concentration, the final surface energy increases with increasing concentration because the increase in the polar component is greater than the decrease in the dispersion component. It can be seen that the contribution of the polar component plays a major role in the surface energy of the W-OH solid consolidation compared to the dispersion component.

The contact angles and standard deviations of the W-OH solid consolidation for each concentration are shown in Table 6, and the maximum standard deviation was found to be 2.03%, which indicates that the reproducibility of the test is good.

#### 3.2.2. Adhesion Model Analysis of the W-OH Solid Consolidation under Dry–Wet Cycles

Because the Pisha sandstone will be thoroughly infiltrated by the W-OH solution and water, it can be approximately considered that the contact angles, $\theta_{SW}$, between the water and the Pisha sandstone and the $\theta_{SL}$ between the W-OH solution and the Pisha sandstone are 0° at that time. The contact angle $\theta_{LW}$ can be measured directly by the contact angle measuring instrument. Besides, because the contact angle of the water on the W-OH solid consolidation changes rapidly, it is believed that the contact angle $\theta_{LW}$ is obtained when the dropping time is 60 s; the value of $\theta_{LW}$ is shown in Table 7. When the surface tension, $\gamma_{W}$, of water is 72.8 ${\mathrm{mJ}/m}^{2}$, then Equations (7), (8), and (10) can be written as: (13)$$W_{SL}=2\lambda_{L}$$
(14)$$\Delta G_{SL}=-2\gamma_{L}$$
(15)$$\Delta G_{SLW}=2\lambda_{L}-72.8\left(1+\cos\theta_{LW}\right)$$

According to Equations (11)–(15), the adhesion work, $W_{SL}$, between W-OH (at various concentrations) and the Pisha sandstone, meaning the specific surface energy change value $\Delta G_{SL}$ in the two-phase system and $\Delta G_{SLW}$ in the three-phase system, can be calculated, as shown in Table 8. The relationship between $\left|\Delta G_{SL}\right|$ and concentration and the relationship between $\left|\Delta G_{SLW}\right|$ and concentration are fitted respectively, as shown in Equations (16) and (17). And it is found that the absolute value for both satisfies the linear function well with the W-OH concentration.
(16)$$\left|\Delta G_{SL}\right|=3.856n+78.618\;R^{2}=0.945$$
(17)$$\left|\Delta G_{SLW}\right|=-2.833n+38.89\;R^{2}=0.905$$

$\Delta G_{SL}$ is a negative value, indicating that the energy is released spontaneously when W-OH and the Pisha sandstone adhere to each other. The greater the absolute value is, the better the adhesion between the two is. With an increase in concentration, the absolute value of $W_{SL}$ and $\Delta G_{SL}$ increases, indicating that the higher the concentration of the W-OH, the better the adhesion between W-OH and the Pisha sandstone. Similarly, the negative value of $\Delta G_{SLW}$ shows that water releases energy when absorbed into the W-OH–Pisha sandstone system, and the water failure process is also spontaneous. Therefore, the greater the absolute value of $\Delta G_{SLW}$, the more seriously W-OH is exfoliated from the Pisha sandstone. With an increase in the W-OH concentration, the smaller the absolute value of $\Delta G_{SLW}$, and the lighter the exfoliation degree.

In conclusion, with an increase in W-OH concentration, we see better adhesion to the Pisha sandstone, with stronger resistance to water failure (antiexfoliation ability), and the better the matching performance of this combination. Some scholars [36] use Equation (18) to represent the matching performance of the asphalt and aggregate combination. Similarly, Equation (18) can also be used to express the matching performance between W-OH and the Pisha sandstone.
(18)$$\gamma =\left|\frac{\Delta G_{SL}}{\Delta G_{SLW}}\right|$$

In the formula, γ is defined as the water stability constant, representing the water stability when matching the W-OH and Pisha sandstone combination. The larger γ is, the better the matching is and the better the resistance to water failure is. Besides, in the process of the dry–wet cycles, the exfoliation of W-OH is less. The value of γ at each concentration is shown in Table 6. It can be found that the value of γ increases with an increase in W-OH concentration. 

#### 3.2.3. Verification of the Failure Characteristics of the W-OH Solid Consolidations under Dry–Wet Cycles

By testing the unconfined compressive strength of the W-OH–Pisha sandstone solid consolidations, we can verify the correctness of the model obtained in the previous section. After nine dry–wet cycles, the dried sample was tested, observing the changes in the unconfined compressive strength of the sample. The changes in the adhesion of W-OH to the Pisha sandstone under dry–wet cycles can be analyzed. Finally, the theoretical analysis of the adhesion failure of the W-OH–Pisha sandstone system under dry–wet cycles is verified based on the obtained results. The unconfined compressive strength of the W-OH solid consolidations under different conditions at the same concentration are compared to each other, and the unconfined compressive strength ratio, *T*, is obtained. The relevant dates are shown in Table 9.

With the same sample preparation material and preparation process, this batch of samples differs only in W-OH concentration. The higher the concentration of W-OH, the more significant the energy change of the two-phase system formed by W-OH and the Pisha sandstone, and the greater the adhesion performance. The stronger the cementation among the particles, the greater the force required to destroy the sliding surface. The strength was completely elevated. During the wetting process of the dry–wet cycles, W-OH shedding from the Pisha sandstone is continuously replaced by water, and the W-OH–Pisha sandstone system becomes the W-OH-water system and the Pisha sandstone-water system due to the replacement of W-OH by water. Moreover, the higher the concentration of W-OH, the greater the energy consumed by the replacement process, with W-OH less likely to be lost and shed. This results in less strength loss in the solid consolidation and a greater unconfined compressive strength ratio, *T*. This is also consistent with the previous theoretical analysis. 

According to the previous analysis, it is found that the greater the energy change during the adhesion process of W-OH (forming a two-phase system with the Pisha sandstone), the higher the strength of the solid consolidation. The greater the energy change in the exfoliation process (W-OH–Pisha sandstone system is transformed into the W-OH-water system and the Pisah sandstone-water system), the less W-OH is replaced by water and is exfoliated, with the strength of the solid consolidation reduced during the dry–wet cycles. Therefore, the strength of the solid consolidation is influenced by the adhesion and exfoliation processes. The correlation between the water stability constant and the unconfined strength ratio is now analyzed; the results are shown in Figure 9.

Fitting for the data points, and the function formula is shown by Equation (19): (19)$$T=65.13+11.6\ln(\lambda -3.31)\;R^{2}=0.90$$

From Equation (19) and Figure 9, it can be found that the unconfined strength ratio is influenced by the water stability constant, and the two have a strong correlation. At the stage of having a small water stability constant, the growth of the unconfined strength ratio is faster, after which the growth of the unconfined strength ratio slows down as the water stability constant increases. At different concentrations of W-OH–Pisha sandstone solid consolidation, the unconfined strength ratio of the solid consolidation increases with an increase in the water stability constant, which is macroscopically expressed as increased resistance to water failure and the dry–wet cycles. Therefore, it can be concluded that the magnitude of the water stability constant reflects the strength of the W-OH–Pisha sandstone solid consolidation in terms of its resistance to dry–wet cycles.

## 4. Conclusions

Pisha sandstone areas have serious soil erosion and are the main source of coarse sediment in the Yellow River, and their ecological problems are extremely prominent. The management of the Pisha sandstone areas is urgent. Combining chemical consolidation with vegetation treatment to manage soil erosion in Pisha sandstone areas by using W-OH materials is currently an ideal slope management measure. A great deal of research has been carried out on this at home and abroad, but there is little research on the failure characteristics of the W-OH–Pisha sandstone system. 

In this paper, we investigate the cohesive failure and the adhesion failure of the W-OH–Pisha sandstone system under dry–wet cycles. Besides, an adhesion model and water damage model of the W-OH–Pisha sandstone system were developed. And unconfined compressive strength testing was carried out to verify the correctness of the adhesion model and water damage model. The results of the study provide a theoretical guide for slope management, from which the following findings were obtained.

(1) The mechanical properties test and the surface in-situ observation test of the W-OH–Pisha sandstone system under dry–wet cycles were carried out to investigate the cohesive failure of W-OH solid consolidation. It was found that the mechanical properties of the solid consolidation are not weakened by dry–wet cycles. It will only make the surface of the solid consolidation rougher, and all the W-OH solid consolidations had excellent resistance to dry–wet cycles at all concentrations.

(2) Concerning the research on the failure characteristics of the asphalt–aggregate system, the surface energy theory was used to analyze the adhesion process and the water damage spalling process of W-OH and Pisha sandstone. It was found that the larger the concentration of W-OH in the adhesion process, the greater the absolute energy change value (in the formation of the W-OH–Pisha sandstone system) and adhesion performance; in the spalling process, the absolute value of the energy change decreases with an increase in W-OH concentration, indicating that the spalling resistance increases with increasing concentration. In addition, the water stability constant was used to express the compatibility of the W-OH and Pisha sandstone combination. The larger the water stability constant, the better the compatibility between W-OH and the Pisha sandstone, with better resistance to water damage also, and less W-OH is exfoliated during the dry–wet cycles.

(3) Based on the unconfined compressive strength test of the W-OH–Pisha sandstone solid consolidation, an equation for the unconfined strength ratio as a function of the water stability constant was established. It was found that the greater the energy change in the adhesion process of W-OH to the Pisha sandstone under the formation of a two-phase system, the higher the strength of the solid consolidation; the greater the energy change in the exfoliation process, the less likely the W-OH is to be replaced by water and dislodged, and the less reduction in the strength of the solid consolidation. This proves that the theoretical analysis of the adhesion of the W-OH–Pisha sandstone system is correct. Besides, the unconfined strength ratio of the solid consolidation increases with an increase in the water stability constant, which is macroscopically reflected in the increased resistance to water damage and the dry–wet cycles.

As there are still relatively few studies on the failure mechanism of the W-OH–Pisha sandstone system, the contents of this paper may be of some relevance to related studies in the same field. The surface energy theory proposed in this paper to study the adhesion properties of the W-OH–Pisha sandstone system concerning the asphalt–aggregate system provides a method for other similar studies. In addition, as only the effects of the dry–wet cycle conditions on the solid consolidation were investigated in this paper, future studies may be able to consider the effects of other factors (freeze–thaw cycles, temperature, etc.) imposed individually or coupled.

## Figures and Tables

**Figure 1 polymers-14-04837-f001:**
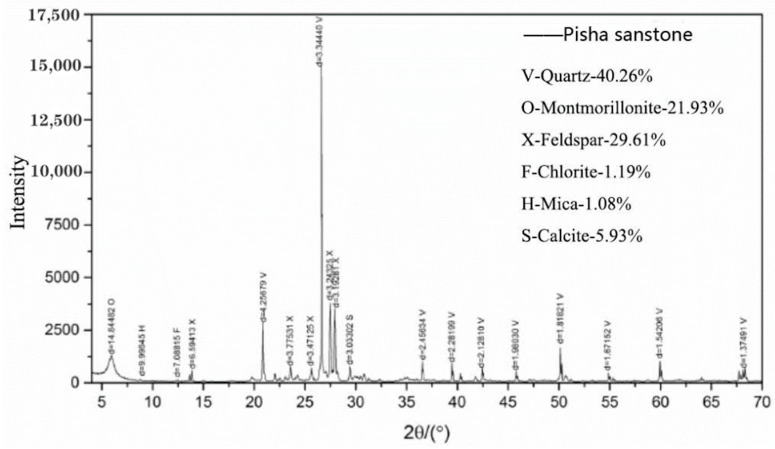
Mineral composition of the Pisha sandstone sample.

**Figure 2 polymers-14-04837-f002:**
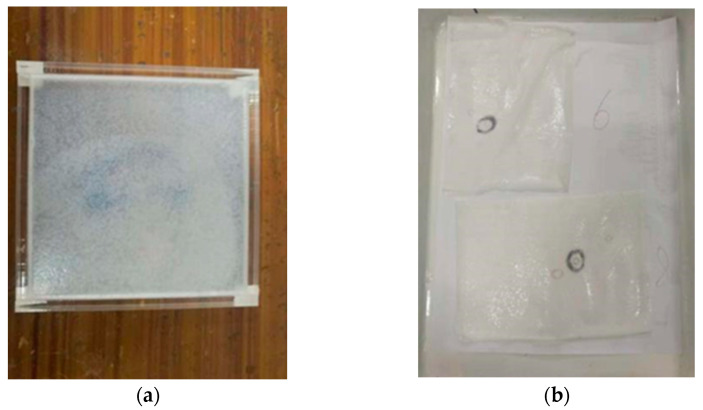
The flow chart of the preparation process of W-OH solid consolidation: (**a**) consolidation into a film; (**b**) consolidation films after demolding.

**Figure 3 polymers-14-04837-f003:**
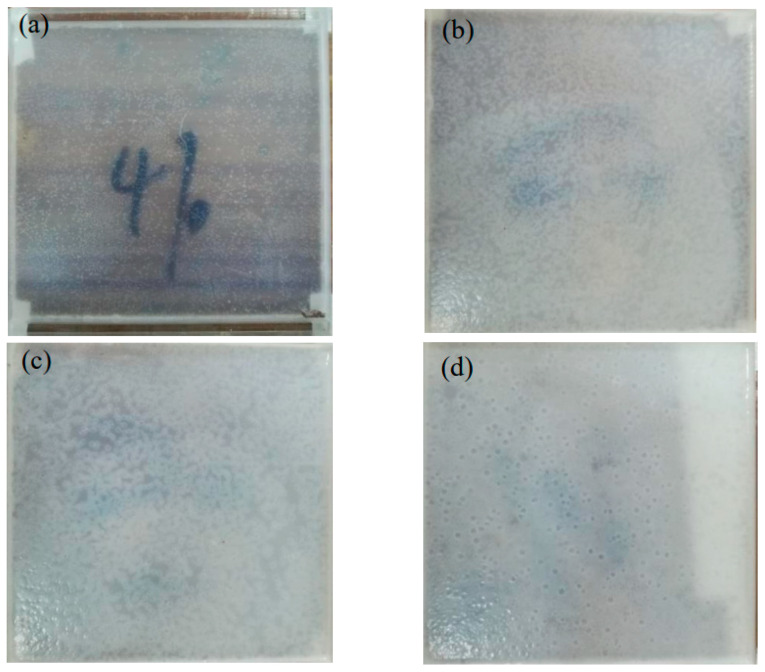
W-OH solid consolidations with different concentrations: (**a**) W-OH concentration: 4%; (**b**) W-OH concentration: 6%; (**c**) W-OH concentration: 8%; and (**d**) W-OH concentration: 10%.

**Figure 4 polymers-14-04837-f004:**
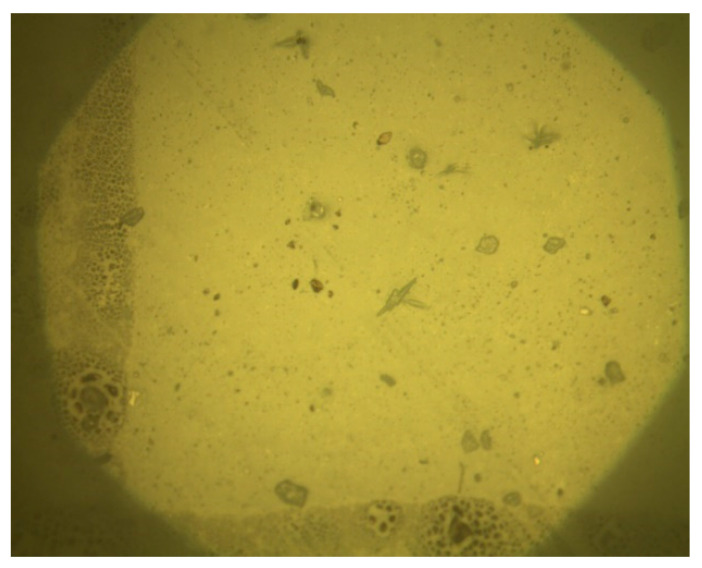
The sample under the metallographic microscope.

**Figure 5 polymers-14-04837-f005:**
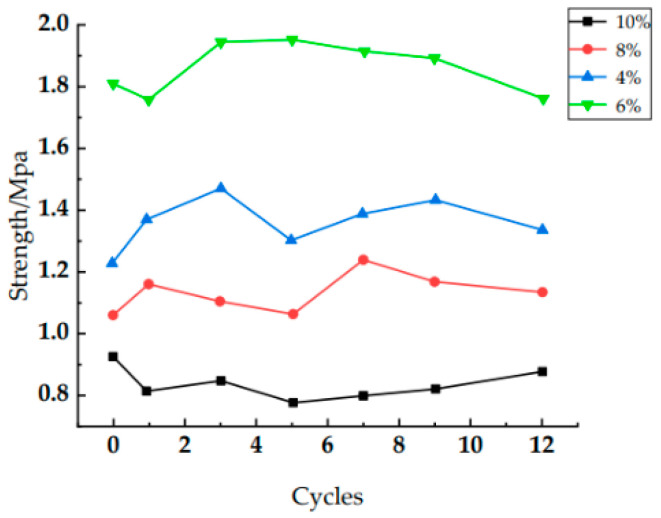
Positioning shift strength of different numbers of dry–wet cycles.

**Figure 6 polymers-14-04837-f006:**
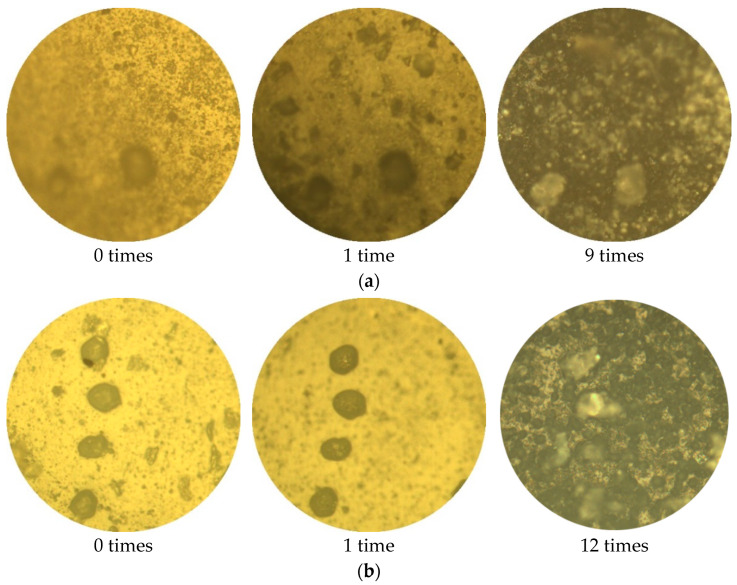
Surface morphology of the W-OH solid consolidation after dry–wet cycles: (**a**) W-OH concentration: 4%; (**b**) W-OH concentration: 6%; (**c**) W-OH concentration; 8%; and (**d**) W-OH concentration: 10%.

**Figure 7 polymers-14-04837-f007:**
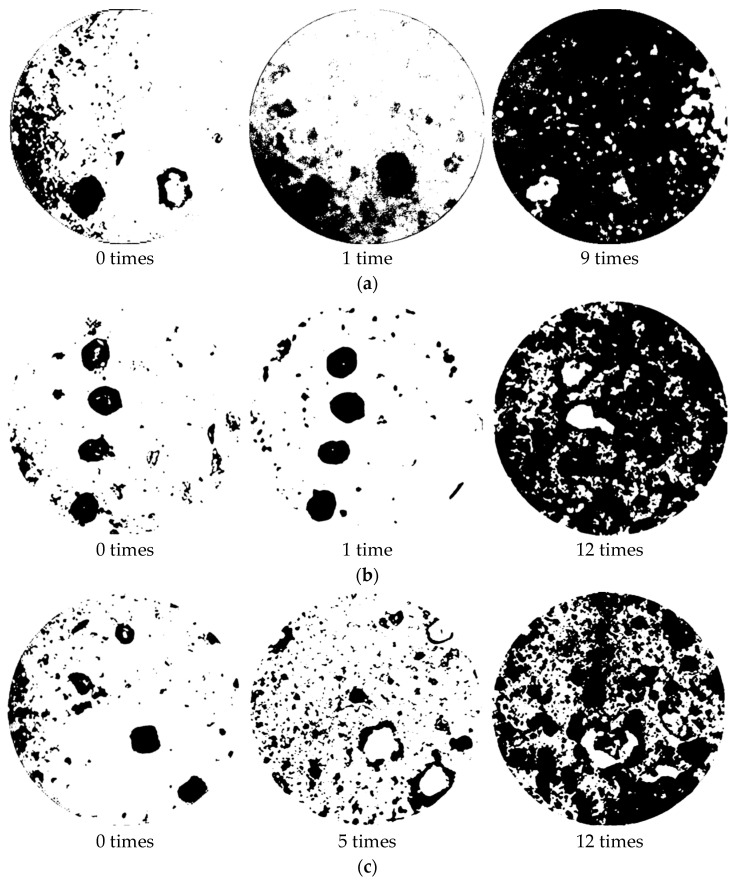
Surface morphology of the W-OH solid consolidation after binarization treatment: (**a**) W-OH concentration: 4%; (**b**) W-OH concentration: 6%; (**c**) W-OH concentration: 8%; and (**d**) W-OH concentration: 10%.

**Figure 8 polymers-14-04837-f008:**
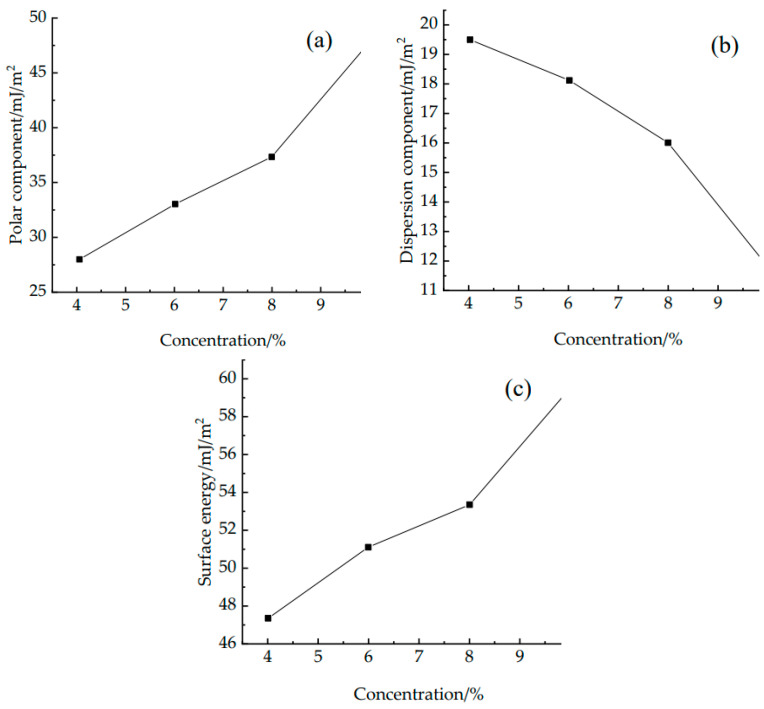
Curves of the surface energy and surface energy parameters with changes in W-OH concentration: (**a**) polar component; (**b**) dispersion component; and (**c**) surface energy.

**Figure 9 polymers-14-04837-f009:**
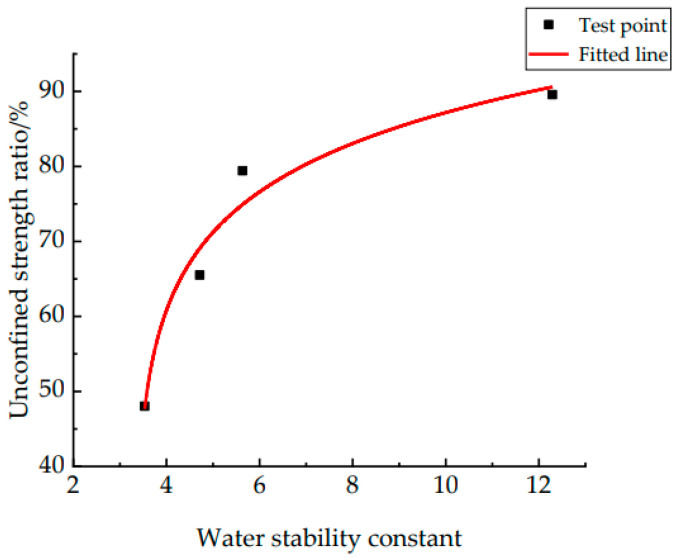
*T* and γ correlation diagram.

**Table 1 polymers-14-04837-t001:** The main materials used in the synthesis of W-OH.

**Main Materials**	Carbodiimide modified MDI ^1^	Toluene diisocyanate (TDI)	Polyether polyol	Butanone
**Source**	Toho Chemical Industry (Shanghai, China)	Bayer (Shanghai) Polymer (Shanghai, China)	Kuraray (Okayama, Japan)	Jusen Chemical (Shanghai, China)

^1^ 4,4′-diphenylmethane diisocyanate.

**Table 2 polymers-14-04837-t002:** Raw material dosage of W-OH films with different concentrations.

Concentration (%)	Water (g)	W-OH (g)
4	28.8	1.2
6	28.2	1.8
8	27.6	2.4
10	27.0	3.0

**Table 3 polymers-14-04837-t003:** The thickness of W-OH solid consolidations using different concentrations.

**Consolidation (%)**	4	6	8	10
**Thickness (mm)**	0.124	0.156	0.325	0.555

**Table 4 polymers-14-04837-t004:** Surface free energy of the test liquid and its components.

Test Liquid	Surface Free Energy		
	$\gamma_{L}$	$\gamma_{L}^{d}$	$\gamma_{L}^{p}$
Formamide	59.0	39.4	19.6
Ethylene glycol	48.3	29.3	19.0

**Table 5 polymers-14-04837-t005:** Variations in dark spot rate with the number of dry–wet cycles.

W-OH Concentration (%)	Number of Dry–Wet Cycles						
	0	1	3	5	7	8	12
4	8.73	18.50	30.10	51.80	65.30	70.20	/
6	8.30	7.60	11.60	16.60	36.10	45.30	61.20
8	9.25	9.12	9.88	22.77	35.02	46.89	49.13
10	14.90	12.50	16.90	27.10	32.8	42.3	44.00

**Table 6 polymers-14-04837-t006:** Contact angle values of plant consolidation with different concentrations.

Concentration (%)	Formamide			Ethylene Glycol		
	Contact Angle (°)	Average Value (°)	Standard Deviation (%)	Contact Angle (°)	Average Value (°)	Standard Deviation (%)
4	43.25	43.00	1.62	42.83	45.13	1.81
	40.42			43.83		
	42.33			45.00		
	43.67			45.92		
	45.33			48.08		
6	38.83	39.89	0.97	43.83	42.85	1.17
	41.15			41.33		
	40.16			41.50		
	40.00			43.83		
	38.92			43.75		
8	41.25	39.85	0.91	45.50	43.51	1.83
	40.58			43.00		
	39.08			40.50		
	39.42			45.33		
	38.92			43.25		
10	40.08	39.95	0.69	46.67	45.07	2.03
	38.92			42.58		
	38.92			47.50		
	40.50			45.83		
	40.33			42.75		

**Table 7 polymers-14-04837-t007:** The contact angle between the water and W-OH at various concentrations.

**Concentration (%)**	4	6	8	10
**Contact Angle (°)**	48.0	45.5	43.5	40

**Table 8 polymers-14-04837-t008:** W-OH at various concentrations and the $W_{SL}$, $\Delta G_{SL}$, and $\Delta G_{SLW}$ of the Pisha sandstone.

Concentration (%)	$W_{SL}$ $\;(mJ/m^{2})$	$\Delta G_{SL}$ $\;(mJ/m^{2})$	$\Delta G_{SLW}$ $\;(mJ/m^{2})$	γ $\;(mJ/m^{2})$
4	94.70	−94.70	−26.81	3.53
6	102.16	−102.16	−21.67	4.71
8	106.68	−106.68	−18.93	5.64
10	118.90	−118.90	−9.67	12.29

**Table 9 polymers-14-04837-t009:** The W-OH–Pisha sandstone solid consolidation: $\sigma_{c}$, and the unconfined strength ratio: *T*.

Concentration (%)	$\sigma_{after\;processing}\;(\mathrm{kPa})$	$\sigma_{untreated}\;(\mathrm{kPa})$	*T* (%)
4	114.83	239.03	48.04
6	461.26	704.04	65.52
8	830.39	1045.65	79.41
10	1487.16	1660.25	89.57

## Data Availability

The data presented in this study are available on request from the corresponding author.

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
