# Peer review of "Study on the Failure Mechanism of a Modified Hydrophilic Polyurethane Material Pisha Sandstone System under Dry–Wet Cycles"

_polymers, 2022, doi:10.3390/polym14224837_

Round 1
Reviewer 1 Report
From my point of view, the paper (and the research) is potentially of interest but English is very difficult to follow (a deep revision by a native speaker is mandatory) and some important keys are missing in the manuscript. My overall recommendation is Reject.
Some points to consider:
- Line 32: Arsenic sandstone? Explain profusely
- Line 53: W-OH only the first time
- Lines 57-59: Mentioned before in the introduction
- Figures 1 and 2 are from other paper [19]?
- Lines 75-77: Mentioned before in the introduction
- Line 77: Xu et al [21] Revise
- Line 83: Littlehe et al [23] Revise
In general, introduction is very confusing. Please, order in a more appropriate form.
- Lines 114-119: A Table with the composition of the plant consolidation materials should be shown
- Lines 122-126: Analysis by XRD? Specify and explain in detail
- Figures 3, 6, 7, 9 and 10 are not necessary
- Lines 275-302: Please explain the behavior of the samples at 4 and 6 %
- Figure 13: For comparative purposes, images should be taken at the same number of cycles
- Lines 328-352: Optical observations should be ratified by using SEM
- Lines 367-379: Formation and breakage of bonds should be followed by using spectroscopic techniques as FTIR and/or Raman. In addition, more discussion are necessary regarding the contact angle measurements
- Table 5 or Figure 15, both are redundant
Reviewer 2 Report
1. English should be checked by professional as there are many grammatical errors.
2. Abstract is not written well and should be rewritten to include quantitative findings.
3. Author should highlight the importance of use of the current approach over earlier approaches
4. What is the novelty of the approach used and how it is different from other methods used wordwide
5.Pease validate the obtained results with field condition
6.Please highlight the innovative component of the work
7. some of the relevant references should be added like
Evolution of thermal damage threshold of Jalore granite PK Gautam, AK Verma, P Sharma, TN Singh - Rock Mechanics and Rock Engineering, 2018 Thermomechanical analysis of different types of sandstone at elevated temperature PK Gautam, AK Verma, S Maheshwar, TN Singh - Rock Mechanics and Rock Engineering, 2016 Experimental investigations on the thermal properties of Jalore granitic rocks for nuclear waste repository PK Gautam, AK Verma, TN Singh, W Hu, KH Singh - Thermochimica Acta, 2019
Reviewer 3 Report
The paper called Study on interface failure mechanism of plant consolidation material-Pisha sandstone under dry-wet cycle by Wenbo Ma, Peng Tang , Xuan Zhou, Guodong Li, and Wei Zhu.
The paper is very good; there are only few small improvements to make. There are some major aspects I would like to highlight. There are some things that could be added to the paper to broaden the scope of the paper along with the group of potential readers.
Very good research work, requires a few additions and corrections;
1) The title should be redrafted to reflect the nature of the work.
2) The Abstract should specify the purpose and scope of the work,
3) In the introduction, a few sentences should be added regarding the state of world research
4) The chapter result and discussion should be separated because it is not known what is what,
5) Conclusions should be extended as far as possible because it is too poor for the rest of the work
6) Please expand your literature review on this topic.
The presented conclusions may be of fundamental importance, therefore they should be presented in a better light and the author(s) should emphasize the original research contribution. I believe, that suggested amendments will significantly increase the relevance of the publication and will improve it. After applying all required changes, the paper is suitable for publication
Round 2
Reviewer 1 Report
Although my previous recommendation was to reject the article, the authors have made a great effort to improve the article and my decission was major revision regarding the new version.
Some items to consider :
- Figures 1 and 2 are from other papers of the authors, can be reproduced in this paper?
- XRD experimental conditions must be provided (source, detector, 2-theta range, steps, etc.)
- Response concerning the behaviour of samples at 4 % and 6 % should be included in the text
- Although other experimental techniques should have been employed, I understand that they are not available for the authors
Reviewer 3 Report
thank you for the corrections made, now the paper is ready for publication
Author Response
It is a great honor to be recognized by you for this work. Your valuable suggestions have greatly improved the quality of the article. Thanks to the reviewers and editors for their help in revising and publishing this article. Wish you a happy life and good health.
Round 3
Reviewer 1 Report
It can be accepted in its current form